# Capsaicin Targets Lipogenesis in HepG2 Cells Through AMPK Activation, AKT Inhibition and PPARs Regulation

**DOI:** 10.3390/ijms20071660

**Published:** 2019-04-03

**Authors:** Alicia Bort, Belén G. Sánchez, Pedro A. Mateos-Gómez, Inés Díaz-Laviada, Nieves Rodríguez-Henche

**Affiliations:** 1Department of Systems Biology, School of Medicine and Health Sciences, University of Alcalá, Alcalá de Henares, E-28871 Madrid, Spain; aliciabort@gmail.com (A.B.); belensg.88@gmail.com (B.G.S.); pedroantonio.mateos@uah.es (P.A.M.-G.); ines.diazlaviada@uah.es (I.D.-L.); 2Chemical Research Institute “Andrés M. del Río” (IQAR), University of Alcalá, Alcalá de Henares, E-28871 Madrid, Spain

**Keywords:** capsaicin, AMPK, AKT, PPARγ, PPARα, HepG2, lipid metabolism

## Abstract

Obesity, a major risk factor for chronic diseases such as type 2 diabetes (T2D), represents a serious primary health problem worldwide. Dietary habits are of special interest to prevent and counteract the obesity and its associated metabolic disorders, including lipid steatosis. Capsaicin, a pungent compound of chili peppers, has been found to ameliorate diet-induced obesity in rodents and humans. The purpose of this study was to examine the effect of capsaicin on hepatic lipogenesis and to delineate the underlying signaling pathways involved, using HepG2 cells as an experimental model. Cellular neutral lipids, stained with BODIPY493/503, were quantified by flow cytometry, and the protein expression and activity were determined by immunoblotting. Capsaicin reduced basal neutral lipid content in HepG2 cells, as well that induced by troglitazone or by oleic acid. This effect of capsaicin was prevented by dorsomorphin and GW9662, pharmacological inhibitors of AMPK and PPARγ, respectively. In addition, capsaicin activated AMPK and inhibited the AKT/mTOR pathway, major regulators of hepatic lipogenesis. Furthermore, capsaicin blocked autophagy and increased PGC-1α protein. These results suggest that capsaicin behaves as an anti-lipogenic compound in HepG2 cells.

## 1. Introduction

Obesity promotes alterations in hepatic lipid and glucose metabolism, which are intimately linked to the development of non-alcoholic fatty liver disease (NAFLD) [1]. NAFLD is characterized by an abnormal accumulation of fat in the liver due to causes other than excessive alcohol intake [2], and is the most common chronic liver disease, affecting one quarter of the global population [3]. Ectopic accumulation of lipids within the liver causes metabolic disturbances that contribute to the development of type 2 diabetes (T2D), cardiovascular disease (CVD), non-alcoholic steatohepatitis (NASH) and hepatocellular carcinoma (HCC) [2,4,5]. However, the association between NAFLD and other metabolic disorders may be bidirectional, since up to 75% of individuals with type 2 diabetes have NAFLD and patients with NAFLD are also at increased risk of developing incident diabetes [6].

Insulin resistance is a hallmark of metabolic diseases, resulting in the impaired ability of insulin to suppress liver gluconeogenesis, while maintaining its ability to stimulate lipogenesis [7]. Therapies that improve insulin resistance also reduce liver steatosis. Metformin and thiazolidinediones (TZDs) act as potent insulin sensitizers and are widely used as antidiabetic agents. Both increase peripheral glucose uptake, decrease fasting fatty acid concentration, increase circulating adiponectin concentration and decrease pro-inflammatory cytokines [8,9]. In addition, metformin reverses fatty liver disease in obese, leptin-deficient mice [10] and has beneficial effects in NAFLD patients [11]. The molecular mechanisms by which metformin exerts its beneficial metabolic effects are complex and still not fully understood. Metformin primarily targets the mitochondrial respiratory-chain complex 1, thus lowering cellular energy charge and leading to AMP-activated kinase (AMPK) activation, which mediates lipid lowering [12,13,14]. However, AMPK may not account for all its actions [14,15]. On the other hand, TZDs act as ligands of the peroxisome proliferator-activated receptor γ, PPARγ, a member of the nuclear receptor superfamily that is essential for adipocyte differentiation, survival and lipid metabolism [16,17]. Although liver PPARγ expression is relatively low in basal conditions, it acts as a steatogenic factor [18] and its expression is upregulated in fatty liver diseases such as NAFLD, where it induces *de novo* lipogenesis and lipid accumulation [19].

The balance between fatty acid uptake, *de novo* lipogenesis, triglyceride synthesis, fatty acid oxidation, and triglyceride export determine the amount of fat stored in the liver. Thus, dysregulation in any of these processes leads to lipid imbalance and fatty acid disease. AMPK, as a major sensor of the energy status, maintains energy homeostasis of the cell by activating catabolic pathways that generate ATP, while inhibiting anabolic pathways that consume ATP [20,21]. AMPK inhibits *de novo* lipogenesis by inducing inhibitory phosphorylation of two direct substrates: acetyl-CoA carboxylase 1 (ACC1) [22], which catalyzes the rate-limiting step in fatty acid synthesis by converting acetyl-CoA to malonyl-CoA, and sterol regulatory element-binding protein 1c (SREBP-1c) [23], transcription factor that regulates the expression of genes involved in fatty acid metabolism [24]. Low AMPK activity has been associated with obesity and with insulin resistance states [25]. Conversely, AKT, a serine/threonine kinase downstream of the insulin receptor, regulates multiple metabolic processes including activation of *de novo* lipogenesis which, in turn, leads to triglyceride accumulation in hepatocytes, hepatic steatosis and a hypertriglyceridemia state [26]. 

Many natural products present in plant-based foods exert beneficial effects on human health and, consequently, they have attracted considerable attention for the management of chronic metabolic diseases. One of these bioactive compounds is capsaicin, the major pungent compound of chili peppers, which ameliorates diet-induced obesity in rodents and humans, supporting a role as an anti-obesity compound [27,28,29]. Dietary capsaicin reduces obesity-related glucose intolerance by regulating inflammatory responses and fatty acid oxidation in obese mice fed with a high-fat diet [30], pointing to its potential for controlling insulin sensitivity and blood glycemia in metabolic disorders. Capsaicin is a selective exogenous agonist for the transient receptor potential vanilloid type 1 (TRPV1), and many of its physiological actions are exerted through TRPV1 activation [31,32,33]. Interestingly, TRPV1 has been identified in metabolically-active tissues, making capsaicin an interesting compound to be tested against metabolic disorders. However, the mechanism by which capsaicin is beneficial in these disorders has not yet been completely unraveled.

In this study, we evaluated the ability of capsaicin to modulate lipid content in HepG2 cells and the underlying signaling pathways involved. The results reported in this study show that capsaicin inhibits *de novo* lipogenesis in HepG2 cells through CaMKKβ/AMPK and PPARγ activation. Capsaicin-induced AMPK activation results in AKT/mTOR and SREBP-1c inhibition, which may be the mechanism by which capsaicin inhibits *de novo* lipogenesis. Our data also reveal that capsaicin blocks autophagy and increases PGC-1α, suggesting enhanced fatty acid oxidation through activation of the PGC-1α-PPARα axis.

## 2. Results

### 2.1. Capsaicin Decreases Neutral Lipid Content in HepG2 Cells

We first analyzed whether capsaicin modifies neutral lipid accumulation in HepG2 cells. To this aim, cells were first serum-starved for 24 h and then incubated with capsaicin for 24 h. The intracellular neutral lipid content was measured by staining cell lipid droplets with BODIPY 493/503 for 30 min. As shown in Figure 1A, doses of 200 µM and 300 µM capsaicin decreased neutral lipid content in HepG2 cells by 20% and 40%, respectively. The dose of 200 µM capsaicin was then chosen to continue the study as it was the lowest dose that produces the effect. We next evaluated endogenous expression of TRPV1, as capsaicin is an agonist of this channel [31]. As shown in Figure 1B, HepG2 cells express endogenous TRPV1 protein and capsaicin induced an increase in its expression as earlier as 1 h after treatment, with a greater effect at 8 h of treatment. Likewise, capsaicin inhibited neutral lipid accumulation and upregulated TRPV1 expression in HepG2 cells exposed to oleic acid (Appendix A). These results indicate that TRPV1 expression is important for the capsaicin action in HepG2 cells. A similar capsaicin effect has been observed in mice epididymal and subcutaneous pre-adipocytes, in which dietary capsaicin counteracted the suppressive effect of a high fat diet on TRPV1 expression [34].

### 2.2. AMPK is Involved in the Capsaicin-Induced Reduction of Neutral Lipid Content in HepG2 Cells

AMPK is the major sensor of the cellular energetic status, blocking cellular anabolism and activating cellular catabolism under ATP depletion and/or metabolic stress. AMPK directly impacts lipid metabolism by inhibiting acetyl-CoA carboxylase (ACC), the enzymatic rate-limiting step in fatty acid synthesis, leading to lipogenesis inhibition and β-oxidation activation [22]. AMPK activation by metformin reduces hepatic steatosis [35] and we have recently documented that capsaicin activates AMPK in HepG2 cells through CaMKKβ [36]. So, we wondered whether AMPK was involved in the capsaicin-induced inhibition of the neutral lipid content in HepG2 cells. First, we compared the effect of capsaicin with that produced by AICAR, a well-established pharmacological AMPK activator. We treated cells with 1 µM AICAR or 200 µM capsaicin for 24 h and then measured neutral lipid content. As shown in Figure 2A, AICAR efficiently diminished neutral lipid content in HepG2 cells, the capsaicin effect being 57% of that exerted by AICAR. To address whether AMPK mediated the capsaicin effect, cells were pretreated for 30 min with 5 µM dorsomorphin, an AMPK inhibitor, and then with 200 µM capsaicin for 24 h. As shown in Figure 2A, dorsomorphin prevented the capsaicin-induced neutral lipid reduction. The same effect was observed when cells were pretreated with STO-609, an CaMKKβ inhibitor. These findings suggest that capsaicin decreases neutral lipid content in HepG2 cells via a CaMKKβ/AMPK-mediated mechanism. Accordingly, and as shown in Figure 2B, capsaicin activated AMPK in HepG2 cells, as shown by the increased phosphorylation of AMPK at Thr172 as well as the corresponding phosphorylation of its downstream target ACC at Ser79, proving *de novo* lipogenesis inhibition. 

We also assessed the status of the sterol regulatory element-binding protein 1c (SREBP-1c), the dominant transcriptional regulator of *de novo* lipogenesis in liver, which is negatively regulated by AMPK [23]. SREBP-1c is synthesized as an inactive precursor that is inserted into the endoplasmic reticulum (ER) membrane, where its C-terminus interacts with SCAP (SREBP cleavage-activating protein). SCAP forms a complex with insulin-induced gene (INSIG) proteins, which retain SREBP-1c in the ER. Under appropriate signals, i.e., insulin, the SREBP-1c-SCAP complex dissociates from INSIG and migrates to the Golgi, where it is activated by proteolytic cleavage. The N-terminal form of SREBP (mature form SREBP-1c or mSREBP-1c) translocates to the nucleus where it promotes the expression of lipogenic genes such as ACC, FASN (fatty acid synthase) or ACLY (ATP citrate lyase) [37]. Activated AMPK phosphorylates SREBP-1c at Ser372, inhibiting its proteolytic processing, thus preventing its nuclear translocation and lipogenic activity [23]. As shown in Figure 2C, capsaicin induced phosphorylation of SREBP-1c at Ser372 and this phosphorylation was prevented by dorsomorphin, suggesting that capsaicin inhibits SREBP-1c through AMPK activation.

### 2.3. PPARγ is Involved in the Capsaicin-Induced Reduction of Neutral Lipid Content in HepG2 Cells

In an attempt to further elucidate the mechanisms underlying the capsaicin-induced decrease of neutral lipid content we assessed PPARγ, a ligand-activated nuclear receptor that functions as a transcriptional factor to regulate lipid metabolism. First, we analyzed PPARγ expression in HepG2 cell lysates treated with 200 µM capsaicin for 8 h. As shown in Figure 3A, capsaicin increased PPARγ protein content, as well as that of the fatty acid transporter CD36, currently CD36/SRB2 as it is known to be a member of the scavenger receptor protein superfamily [38]. CD36 facilitates the transport of fatty acids across the plasma membrane and also transduces intracellular signaling events that influence how the fatty acids are utilized, thus coordinating fatty acid uptake with its intracellular fate [39]. We then evaluated whether PPARγ was involved in the capsaicin-induced reduction of neutral lipids. Thus, HepG2 cells were pretreated for 30 min with 3 µM GW9662, a PPARγ-specific inhibitor, and then with 200 µM capsaicin for 24 h. As shown in Figure 3B, GW9662 alone did not modify basal neutral lipid content whereas in combination with capsaicin, GW9662 prevented the capsaicin effect, indicating that PPARγ participates in the mechanism underlying the capsaicin-induced reduction of neutral lipids in HepG2 cells. To further determine the role of PPARγ, we activated PPARγ with its well-known agonist troglitazone, the first thiazolidinedione approved as an antidiabetic agent for its insulin sensitizer action [40]. HepG2 cells were pretreated with 10 µM troglitazone and then with 200 µM capsaicin for 24 h. As shown in Figure 3C, capsaicin significantly inhibited troglitazone-induced neutral lipid content. This result further reinforces the idea that capsaicin inhibits neutral lipid accumulation in HepG2 cells and that PPARγ is involved in this effect. In order to know a cross-talk between PPARγ and AMPK, we evaluated whether GW9662 regulates AMPK activity. Our results show that GW9662 does not modify basal or capsaicin-induced AMPK or ACC phosphorylation (Appendix A), indicating that the involvement of PPARγ in the anti-lipogenic effect of capsaicin is not through AMPK activation, and may occur either downstream or be independent of AMPK.

### 2.4. Capsaicin Blocks Autophagy and Upregulates PGC-1α in HepG2 Cells

We then evaluated the impact of capsaicin in autophagy, a catabolic process induced during starvation. Autophagy plays an essential role in cellular homeostasis by channeling cytoplasmic material, including lipid droplets, to degradation within lysosomes [41,42]. So, we asked ourselves whether this catabolic process might be involved in the capsaicin-induced decrease of neutral lipid content in HepG2 cells. To monitor autophagy, we measured two of the autophagy markers, LC3 (microtubule associated protein 1 light-chain 3, a homolog of ATG8) and SQSTM1/p62 (sequestosome 1 or p62) by immunoblotting. LC3 is cleaved and modified by lipidation to generate LC3II, which associates with autophagosome membranes. Thus, the level of LC3II correlates with the number of autophagosomes. LC3 functions as an adaptor protein to recruit selective cargo to the autophagosome via interaction with cargo receptors such as p62. p62 binds and recruits ubiquitinated proteins to autophagosomes on the basis of an ubiquitin-associated (UBA) domain and an LC3-binding domain. p62 itself is degraded in the autophagosome, so p62 accumulates when autophagy is inhibited, and decreased levels can be observed when autophagy is induced. Consequently, it is used to study autophagy flux. As shown in Figure 4A, capsaicin induced the appearance of LC3II, the membrane-associated form of LC3, indicative of autophagy activation. However, the levels of p62 increased, pointing to p62 accumulation and autophagy blockage. It is worth noting the increased levels of LC3I, which itself is also degraded by autophagy, reinforcing the notion that capsaicin induces autophagy blockage. Furthermore, pharmacological inhibition of autophagy by chloroquine, which blocks the final step of autolysosomal degradation, resulted in a slight enhancement of LC3II and p62 accumulation in capsaicin treated cells (Appendix A) consistent with autophagy blockage. These results are in good agreement with our previous results in HepG2 cells and in prostate cancer cells [36,43]. However, GW9662 reduced the capsaicin-induced accumulation of LC3 and p62, indicative of a slight autophagic flux improvement. These results indicate that the capsaicin-induced decrease of neutral lipid content in HepG2 cells is not mediated by autophagy activation. 

Interestingly, specific adipose tissue from autophagy-deficient mice showed increased levels of brown adipogenic factor PPAR-γ transcriptional coactivator (PGC-1 α) and higher adipose tissue β-oxidation rates [44]. Thus, we assessed whether capsaicin regulated PGC-1α in HepG2-cells. As shown in Figure 4B, capsaicin significantly upregulated the expression of this coactivator, pointing to capsaicin activation of the PGC-1α-PPARα axis. We also assessed the expression of CD36, which is reported to be activated by PPAR ligands in a PPAR subtype- and tissue-specific manner [45,46]. We show that capsaicin increased CD36 protein levels, which is in line within PGC-1α-PPARα axis activation. Neither dorsomorphin nor GW9662 prevented capsaicin-induced PGC-1α nor CD36 increase, which suggests that capsaicin upregulates them through an AMPK- and PPARγ- independent mechanism. We then evaluated PPARα expression in HepG2 cells. As shown in Figure 4B, PPARα is expressed in HepG2 cells and capsaicin in combination with GW9662 resulted in PPARα upregulation. These data are consistent with the notion that capsaicin may also decrease neutral lipid content by activating fatty acid transport and oxidation through PGC-1α-PPARα axis upregulation.

### 2.5. Capsaicin Inhibits AKT/mTOR Pathway in HepG2 Cells

Based on the previous results, we evaluated the impact of capsaicin on the AKT/mTOR pathway since it is the major inducer of *de novo* lipogenesis [47] and counteracts the effects of AMPK on the regulation of cellular metabolism, including lipid metabolism [48,49]. In response to insulin and/or nutrient accumulation, AKT is activated by phosphorylation at Thr308 by PDK1 and at Ser473 by mTORC2. Once fully activated, AKT regulates multiple metabolic processes including activation of *de novo* lipogenesis by activating SREBP-1c, through mTORC1-dependent and independent pathways [50], and by inhibiting AMPK through an inhibitory phosphorylation of AMPK at Ser485 [49]. On the other hand, under energetic depletion, AMPK activation inhibits mTORC1 by direct phosphorylation of TCS2 and RAPTOR. As shown in Figure 5A, capsaicin induced a dramatic reduction of phosphorylated AKT at Ser473 and that of its downstream targets mTOR at Ser2448 and S6 at Thr389. Furthermore, dorsomorphin partially prevented the capsaicin-induced AKT/mTOR inhibition, indicating that capsaicin, via AMPK activation, inhibits, at least in part, the AKT/mTOR pathway. These results indicate that capsaicin may inhibit SREBP-1c directly, by activating AMPK, and indirectly, by inhibiting the AKT/mTOR pathway, leading to inhibition of *de novo* lipogenesis. 

Lastly, we investigated whether PPARγ is involved in the capsaicin-induced inhibition of the AKT/mTOR pathway. GW9662 did not modify the capsaicin-induced inhibition of AKT, whereas, it prevented capsaicin-induced mTOR inhibition (Figure 5B). This is in concordance with the fact that GW9662 abrogated the anti-lipogenic effect of capsaicin, and suggests that capsaicin activates PPARγ, leading to mTOR inhibition and *de novo* lipogenesis activation in HepG2 cells. A proposed model underlying molecular mechanisms by which capsaicin reduces neutral lipid content in HepG2 cells in shown in Figure 6. 

## 3. Discussion

Obesity is a major risk factor for metabolic diseases and represents a serious primary health problem worldwide [51]. Overfeeding and a sedentary life style are major contributors of this disorder. Hence, dietary habits are of special interest to prevent and counteract obesity and its associated metabolic complications. Accumulating evidence has shown that dietary capsaicin displays anti-obesity activity by targeting different organs and tissues in the whole body, thus counteracting obesity and the related metabolic disorders such as insulin resistance and liver steatosis [27]. In this study, we evaluated the impact of capsaicin on hepatic lipid accumulation and delineated the cellular signaling pathways involved. We show that capsaicin reduces the neutral lipid content in HepG2 cells through AMPK and PPARγ activation, and increases TRPV1, PPARγ and CD36 protein levels. Furthermore, capsaicin inhibits the AKT/mTOR pathway and SREBP-1c, pointing to lipogenesis inhibition. In addition, capsaicin blocks autophagy and increases PGC-1α protein levels, pointing to activation of the PGC-1α-PPARα axis and fatty acid oxidation in HepG2 cells. 

mTORC1 is key for the adaptive switch between catabolic and anabolic states, and its tight and reciprocal regulation by AMPK and AKT counteracts hepatic steatosis. AMPK maintains the energy homeostasis of the cell by switching on catabolic pathways that generate ATP, while switching off anabolic pathways that consume ATP [20]. In this study, we show that capsaicin-activated AMPK inhibits ACC, the enzyme which generates malonyl-CoA, the FASN substrate for fatty acid synthesis, and SREBP-1c, the transcription factor that controls expression of lipogenic genes such as FAS or ACC, leading to lipogenesis inhibition. Capsaicin also inhibits AKT and mTORC1, which promotes lipogenesis by regulating SREBP-1c [50]. AKT activates SREBP-1c by reducing the expression of the SREBP-1c inhibitor *Insig2a*, facilitating SREBP-1c processing and stabilizing its mature form by blocking GSK3β-mediated degradation [52,53]. It should be interesting to evaluate whether capsaicin regulates the expression of SREBP-1, analyzing *Insig2a* or SCAP expression, and SREBP-1c activity, which would provide insight into a direct functional link between capsaicin-induced AMPK and AKT regulation and SREBP-1 activity. Kim et al. [54] have recently reported that liver *de novo* lipogenesis inhibition by a specific ACC inhibitor reduces hepatic steatosis and insulin resistance in mice. In humans, however, it reduced hepatic steatosis but resulted in hypertriglyceridemia due to activation of SREBP-1c and increased VLDL secretion [54]. Hence, the ability of capsaicin to inhibit both ACC and SREBP-1c might be an advantage to counteract hepatic steatosis and hypertriglyceridemia.

We also evaluated the effect of capsaicin on PPARγ, the master regulator for adipogenesis, fat storage and glucose and lipid metabolism in adipose tissue [16,17], with steatogenic properties in liver [18]. Although liver expression of PPARγ is relatively low in basal conditions, its expression is upregulated in fatty liver diseases such as NAFLD, where it induces *de novo* lipogenesis and lipid accumulation [19]. In this study, we show that capsaicin reduces both oleic acid- and troglitazone-induced neutral lipid levels in HepG2 cells, which demonstrates its anti-lipogenic activity. Capsaicin inhibition of SREBP-1c might be the underlying mechanism involved in the anti-lipogenic effect of capsaicin in HepG2 cells. These data are in good agreement with previous research reporting reduction of hepatic steatosis by dietary capsaicin [30]. We have also shown that capsaicin increases PPARγ protein levels and its inhibition with GW9662 prevents mTOR inhibition and *de novo* lipogenesis inhibition, proving that capsaicin activates PPARγ. Cross-talk between AMPK and PPARs has been previously reported [55]. Nevertheless, in a previous study, we demonstrated that AMPK and PPARγ are two independent pathways activated by cannabinoids [56]. The fact that GW9662 prevents capsaicin-induced mTOR inhibition further confirms the essential role for mTOR in the underlying mechanism for capsaicin’s anti-lipogenic effect. 

We have also analyzed the status of autophagy in the context of HepG2 metabolism. We show that capsaicin induces an autophagy blockage which correlates with upregulation of the PGC-1α protein, a transcriptional coactivator that coordinates mitochondrial biogenesis and energy expenditure [57]. Interestingly, it has been reported that selective adipose tissue autophagy deficiency in mice impairs white adipose tissue differentiation and enhances the expression of brown adipogenic proteins, such as PGC-1α [44]. In addition, dietary capsaicin induces browning of white adipose tissue and upregulates PGC-1α in mice [34]. Therefore, capsaicin-induced PGC-1α expression in HepG2 points to energy expenditure activation. In liver, hepatic PPARα is essential for fatty acid catabolism [58]. However, PPARα alone is not sufficient to induce the expression of target genes, being essential to the participation of the peroxisome proliferator-activated receptor γ coactivator 1α (PGC-1α) [59]. PGC-1α-PPARα complex regulates a set of genes involved in fatty acid transport and β-oxidation [45,60], which are essential for fatty acid catabolism in the fasted state [58]. The PGC-1α-PPARα axis is regulated by the feed-fast cycle. In the feeding state, PPARα is inactivated by the mTORC1-activated nuclear receptor co-repressor 1 (nCoR1)- ribosomal protein S6 kinase 2 (S6K2) complex [61,62]. Thus, capsaicin could induce PGC-1α-PPARα axis activation through PPARα derepression.

The strong inhibition of AKT exerted by capsaicin in HepG2 cells is particularly noteworthy. AKT inhibition is a hallmark of insulin resistance, characterized by the failure of insulin to promote glucose uptake by the muscle and to inhibit gluconeogenesis in the liver. TRIB3, a mammalian homolog of Drosophila tribbles, is a pseudokinase that binds and inhibits AKT, disrupting insulin signaling [63]. Nutrient starvation induces liver TRIB3 expression [64] and overexpression of TRIB3 has been implicated in insulin resistance in animal models as well as in humans [65]. Furthermore, the *TRIB3* promoter contains a PPAR response element (PPRE) and is activated by PPARα in the liver. Therefore, PGC-1α promotes liver insulin resistance through PPARα-dependent induction of TRIB3 [66]. Although we have not measured TRIB3 expression levels, we reasoned that capsaicin-induced PGC-1α-PPARα activation might induce TRIB3 expression in HepG2 cells leading to AKT inhibition. Additional work must be done to confirm this hypothesis. In line with this notion, it has been recently reported that capsaicin promotes apoptosis in cancer cells via TRIB3 upregulation [67]. 

The results reported in this study show that capsaicin inhibits *de novo* lipogenesis in HepG2 cells through CaMKKβ/AMPK and PPARγ activation. Capsaicin-induced AMPK activation results in AKT/mTOR and SREBP-1c inhibition, which may be the mechanism by which capsaicin inhibits *de novo* lipogenesis. Our data also reveal that capsaicin blocks autophagy and increases PGC-1α, suggesting enhanced fatty acid oxidation occurs through activation of the PGC-1α-PPARα axis. 

## 4. Materials and Methods

### 4.1. Reagents

Capsaicin (CAP), AICAR, dorsomorphin (also named compound C), troglitazone and GW9662 were obtained from Tocris (Ellisville, MO, USA). BODIPY (493/503) and STO-609 (a CaMKKβ inhibitor) were from Sigma (St. Louis, MO, USA). Anti-pAMPKα1-Thr172, anti-AMPK, anti-pACC-Ser79, anti-ACC, anti-pSREBP-1c-Ser372, anti-pAKT-Ser473, anti-AKT, anti-pmTOR-Ser2248, anti-mTOR, anti-pS6-Thr389, anti-S6, anti-p62 from Cell Signaling Technology (Danvers, MA, USA), anti-SREBP, anti-CD36, anti-PPARα from Abcam (Cambridge, UK), anti-PPARγ from Santa Cruz Biotechnology (Dallas, TX, USA), anti-PGC-1α, anti-LC3B from Novus Biologicals (Abingdon, UK) and anti-β-tubulin from Sigma (St. Louis, MO, USA).

### 4.2. Cell Culture and Treatments

The human hepatocellular carcinoma HepG2 cell line was purchased from the American Type Culture Collection (ATCC HB-8065, Rockville, MD, USA). Cells were grown in DMEM medium supplemented with 10% fetal bovine serum (FBS), 1% non-essential amino acids and 100 IU/mL penicillin G, 100 mg/mL streptomycin sulfate and 0.25 mg/mL amphotericin B (Invitrogen, Paisley, UK) at 37 °C in a humidified atmosphere with 5% CO_2_. For all treatments, cells were cultured in EMEM medium without serum.

### 4.3. Neutral Lipid Content Measurement

HepG2 cells were seeded in 6-well plates (300.000 cells/well) in a final volume of 1 mL of EMEM. At 24 h following seeding, the medium was aspirated and replaced with fresh EMEM medium devoid of serum and incubated with or without 200 µM capsaicin for 24 h. Inhibitor-pretreatments were added to cells 30 min prior to capsaicin whereas control cells received DMSO alone. Cellular lipid droplets were stained with BODIPY (493/503), a cell permeable lipophylic fluorescence dye that emits bright green fluorescence [68]. BODIPY (493/503) was diluted in DMSO at a concentration of 5 ng/mL and added to the cells 30 min before the end of the treatments. Cells were harvested by trypsin treatment, centrifuged at 500 × *g* for 5 min and then resuspended in 3 mL PBS containing 1.6 μg/mL propidium iodide (Invitrogen, Eugene, OR, USA). Fluorescence intensity was measured in an FACSCalibur flow cytometry system (BD Biosciences, San Jose, CA, USA) and data were analyzed using Cyflogic software V1.2.1 (Perttu Terho, Mika Korkeamaki, CyFlo Ltd., Turku, FINLAND). A total of 5 × 10^3^ events were collected for each sample.

### 4.4. Immunoblotting

Cells were harvested and proteins were extracted using lysis buffer (50 mM Tris pH 7.5, 0.15 M NaCl, 50 mM NaF, 5 mM EDTA, 1 mM EGTA 0.1 mM Na3VO4 0.1% Triton X-100) containing Protease Inhibitor and Phosphatase inhibitor Cocktail (Roche, Diagnostics; Mannheim, Germany), incubated on ice for 20 min and cleared by microcentrifugation. Protein concentrations of cellular lysates were measured by the BioRad™ protein assay kit (Richmond, CA, USA). Equal amounts of protein (20 μg) were loaded in each lane with loading buffer containing 25% 0.25 M Tris-HCl, pH 6.8, 50% glycerol, 10% SDS, 5% mercaptoethanol and 0.025% bromophenol blue. Samples were boiled for 5 min before being separated on 8–10% SDS-PAGE gels, depending on the protein to be analyzed. After electrophoresis, proteins were transferred to polyvinylidene difluoride membranes (BioRad) using an electrophoretic transfer system (Bio-Rad, Hercules, CA, USA). The membranes were then incubated overnight at 4 °C with specific primary antibodies. After washing to remove unbound antibodies, horseradish peroxidase-linked goat anti-mouse or goat anti-rabbit IgG secondary antibodies were added at a dilution ratio of 1:5000 and the membranes were incubated at room temperature for 2 h. The immune complex was visualized with an ECL system (Cell Signaling Technology, Danvers, MA, USA). 

### 4.5. Statistical Analyses

GraphPad Prism 6 software (GraphPad software Inc., la Jolla, CA, USA) was used to determine statistical significances. Results are represented as the mean ± SEM. The data were analyzed using the Student’s t test, one-way ANOVA and Tukey’s multiple comparisons test or two-way ANOVA and Dunnett’s multiple comparisons test. *p* < 0.05 was considered statistically significant.

## Figures and Tables

**Figure 1 ijms-20-01660-f001:**
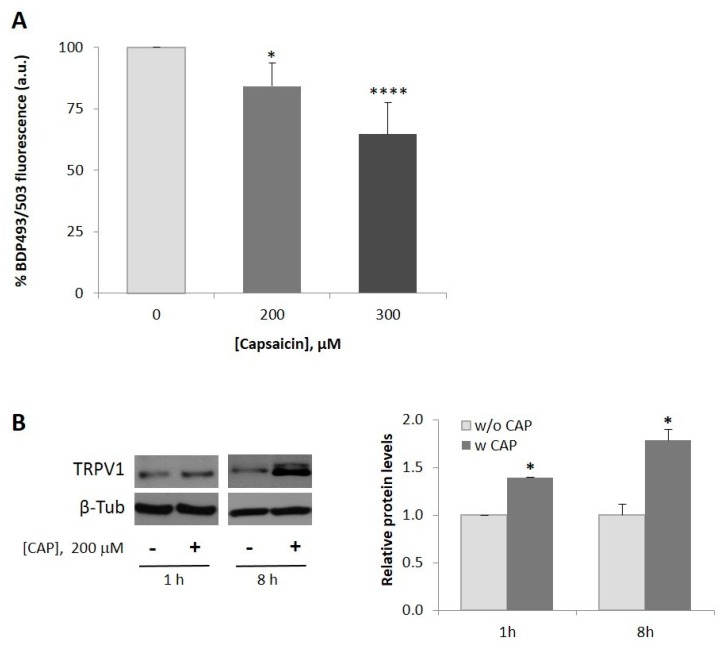
Capsaicin reduces neutral lipid content in HepG2 cells. HepG2 cells were cultured in EMEM medium without serum for 24 h. (**A**) Measurement of neutral lipid content in HepG2 cells. Cells were treated with or without 200 µM or 300 µM capsaicin for 24 h. Intracellular neutral lipids were stained with bodipy 493/503 (BDP 493/503) for 30 min and fluorescence was measured by flow cytometry to quantify neutral lipid content. The fluorescence intensity of untreated cells (0) was used as the control and the rest of the values were expressed as a % of the control value. Results are the mean ± SEM of four experiments. Statistical significance was determined by one-way ANOVA and Tukey’s multiple comparisons test. (**B**) Capsaicin upregulates expression of TRPV1 protein. Cells were treated with or without 200 µM capsaicin for 1 h or 8 h. TRPV1 levels were determined by immunoblotting. β-tubulin was used as a loading control. The densitometric analysis of the bands is shown on the right of the blot. Results are the mean ± SEM of three experiments. Statistical significance was determined by the Student’s t test. Comparisons were vs. untreated cells (* *p* < 0.05, **** *p* < 0.001).

**Figure 2 ijms-20-01660-f002:**
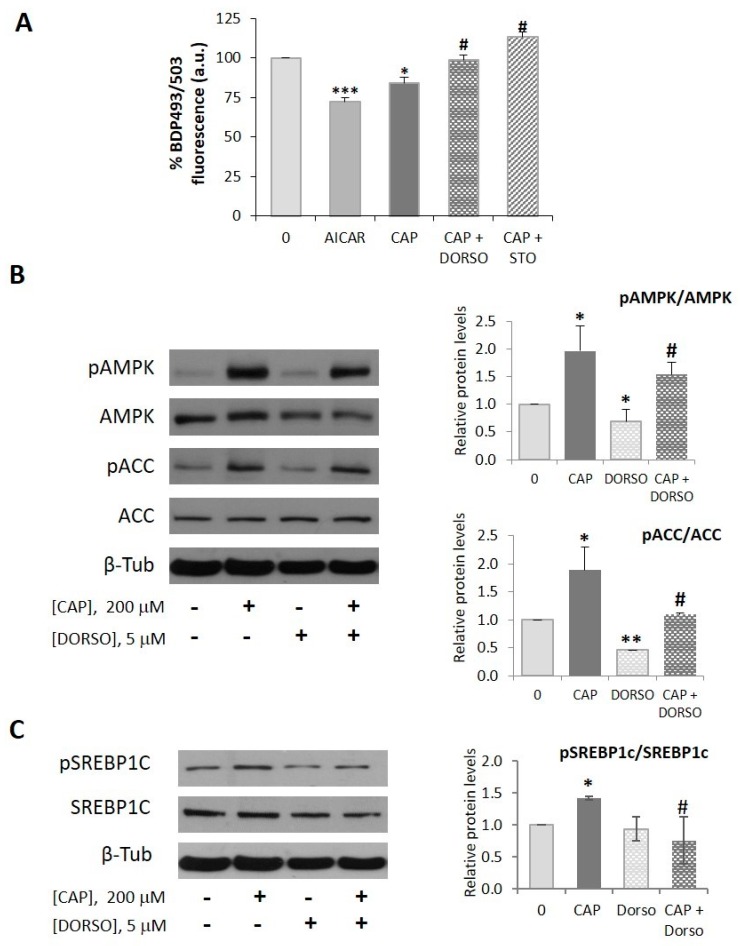
AMPK is involved in the capsaicin-induced reduction of neutral lipid content in HepG2 cells. HepG2 cells were cultured in EMEM medium without serum for 24 h. (**A**) Capsaicin reduces neutral lipid content through CaMKKII/AMPK activation. Cells were pretreated with vehicle, 5 µM dorsomorphin (DORSO) or 10 µM STO-609 (STO) for 30 min and then treated with or without 1 µM AICAR or 200 µM capsaicin (CAP) for 24 h. Intracellular neutral lipids were stained with bodipy 493/503 (BDP 493/503) for 30 min and fluorescence was measured by flow cytometry to quantify neutral lipid content. The fluorescence intensity of untreated cells (0) was used as the control and the rest of values were expressed as % of the control value. Results are the mean ± SEM of four experiments. Statistical significance was determined by one-way ANOVA and Tukey’s multiple comparisons test. (**B**,**C**) Capsaicin activates AMPK. Cells were pretreated with vehicle or 5 µM DORSO, and then treated with or without 200 µM CAP for 1 h. The levels of pAMPK, pACC, pSREBP and their corresponding not phosphorylated forms were determined by immunoblotting. β-tubulin was used as a loading control. The densitometric analysis of the bands is shown on the right. Data are the mean ± SEM of four experiments. Statistical significance was determined by two-way ANOVA and Dunnett’s multiple comparisons test. Comparisons were vs untreated cells (* *p* < 0.05, ** *p* < 0.01, *** *p* < 0.005) or capsaicin treated cells (# *p* < 0.05).

**Figure 3 ijms-20-01660-f003:**
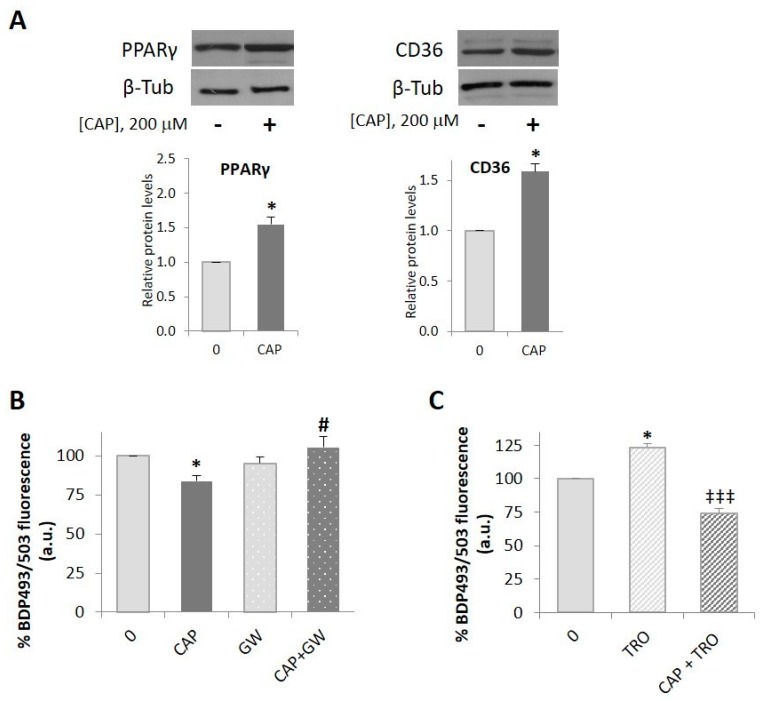
PPARγ is involved in the capsaicin-induced reduction of neutral lipid content in HepG2 cells. HepG2 cells were cultured in EMEM medium without serum for 24 h. (**A**) Capsaicin upregulated PPARγ and CD36 protein expression. Cells were pretreated with vehicle or 3µM of GW9662(GW) for 30 min and then treated with or without 200 µM capsaicin for 8 h. Levels of PPARγ and CD36 were determined by immunoblotting. β-tubulin was used as a loading control. The densitometric analysis of the bands is shown under the blots. Results are the mean ± SEM of three experiments. Statistical significance was determined by the Student’s t test. (**B**) Capsaicin reduced basal neutral lipid content in HepG2 cells. Cells were pretreated with vehicle or 3 µM of GW9662 for 30 min and then treated with or without 200 µM capsaicin (CAP) for 24 h. (**C**) Capsaicin inhibited troglitazone-induced neutral lipid accumulation in HepG2 cells. Cells were pretreated with vehicle or 10 µM of troglitazone (TRO) for 30 min and then treated with or without 200 µM capsaicin (CAP) for 24 h. Intracellular neutral lipids were stained with bodipy 493/503 (BDP 493/503) for 30 min and fluorescence was measured by flow cytometry to quantify neutral lipid content. The fluorescence intensity of untreated cells (0) was used as the control and the rest of values were expressed as % of the control value. Results are the mean ± SEM of four experiments. Statistical significance was determined by one-way ANOVA and Tukey’s multiple comparisons test. Comparisons were vs. control (* *p* < 0.05), capsaicin-treated cells (# *p* < 0.05), or troglitazone-treated cells (‡‡‡ *p* < 0.005).

**Figure 4 ijms-20-01660-f004:**
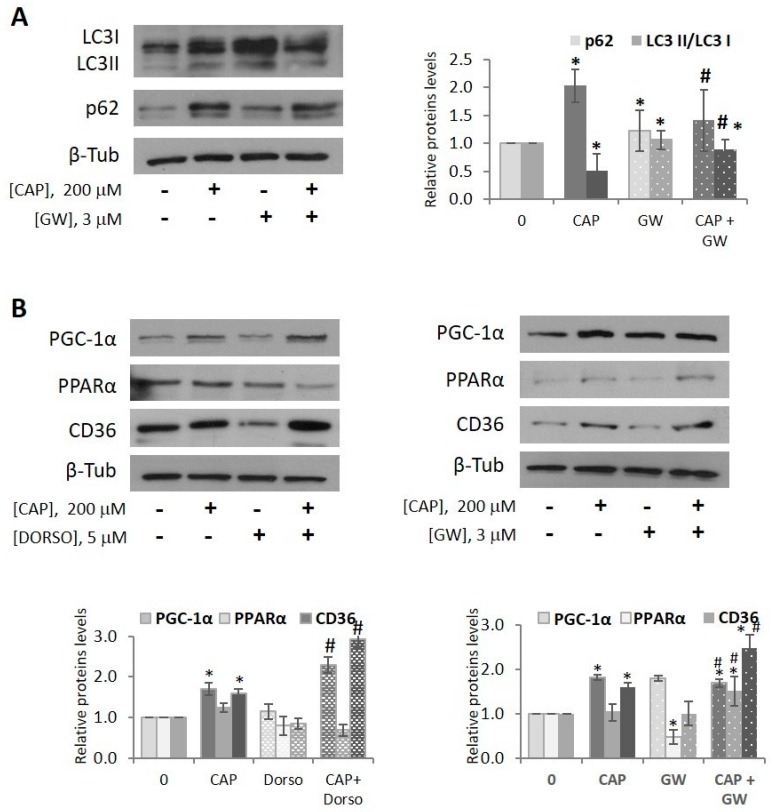
Capsaicin blocks autophagy and upregulates PGC-1α in HepG2 cells. HepG2 cells were cultured in EMEM medium without serum for 24 h. (**A**) Cells were pretreated with vehicle or 3 µM GW9662 (GW) for 30 min and treated with or without 200 µM CAP for 8 h. Cells were lysed and levels of LC3 and p62 were determined by immunoblotting. β-tubulinserves as a loading control. The densitometric analysis of the bands is shown on the right of the blot. (**B**) Cells were pretreated with vehicle, 5 µM dorsomorphin (DORSO) or 3 µM GW9662 (GW) for 30 min and treated with or without 200 µM capsaicin for 8 h. Cells were lysed and levels of PGC-1α, PPARα and CD36 proteins were determined by immunoblotting. β-tubulin serves as a loading control. The densitometric analysis of the bands is shown under the blots. Data are the mean ± SEM of three experiments. Statistical significance was determined by two-way ANOVA and Dunnett’s multiple comparisons test. Comparisons were vs untreated cells (* *p* < 0.05) or capsaicin-treated cells (# *p* < 0.05).

**Figure 5 ijms-20-01660-f005:**
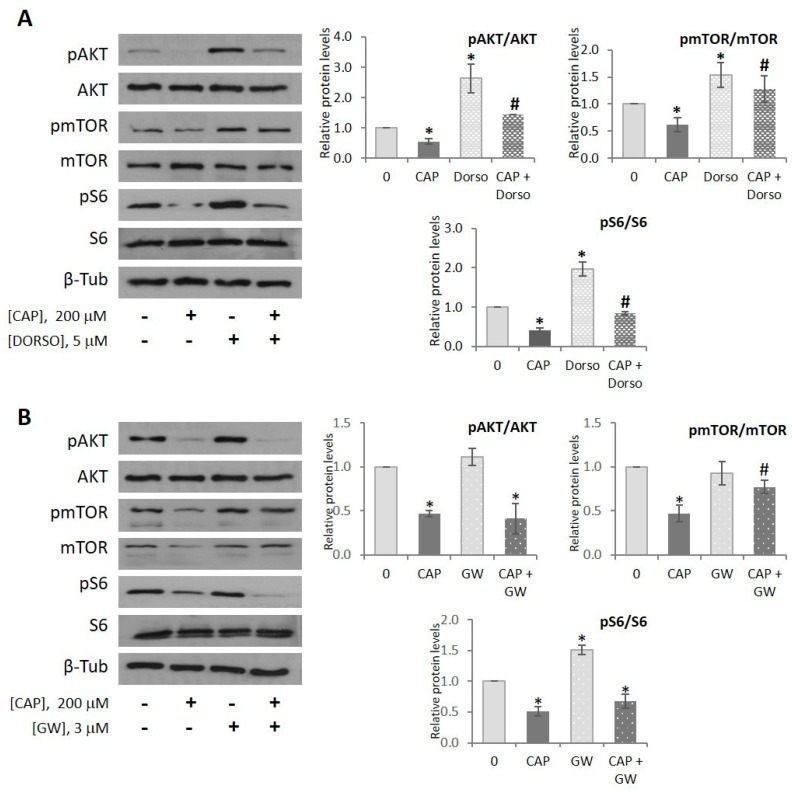
Capsaicin inhibits the AKT/mTORC1 pathway in HepG2 cells. HepG2 cells were cultured in EMEM medium without serum for 24 h. (**A**) Cells were pretreated with vehicle or 5 µM dorsomorphin (DORSO) for 30 min and treated with or without 200 µM CAP for 1 h. Cells were lysed and levels of pAKT (Ser473), pmTOR (Ser2448), pS6 (Thr389) and their corresponding total proteins were determined by immunoblotting. β-tubulin serves as a loading control. The densitometric analysis of the bands is shown on the right. (**B**) Cells were pretreated with vehicle or 3 µM GW9662 (GW) for 30 min and treated with or without 200 µM CAP for 1 h. Cells were lysed and the levels of pAKT (Ser473), pmTOR (Ser2448), pS6 (Thr389) and their corresponding total proteins content were determined by immunoblotting. Data are the mean ± SEM of three experiments. Statistical significance was determined by two-way ANOVA and Dunnett’s multiple comparisons test. Comparisons were vs untreated cells (* *p* < 0.05) or capsaicin-treated cells (# *p* < 0.05).

**Figure 6 ijms-20-01660-f006:**
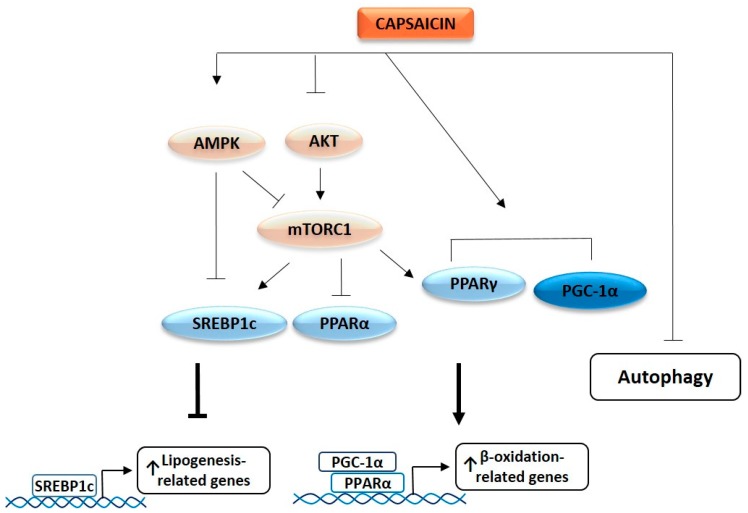
A model showing the proposed molecular mechanism by which capsaicin reduces neutral lipids in HepG2 cells.

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
