# Peer review of "Capsaicin Targets Lipogenesis in HepG2 Cells Through AMPK Activation, AKT Inhibition and PPARs Regulation"

_ijms, 2019, doi:10.3390/ijms20071660_

Round 1

Reviewer 1 Report

This is an interesting manuscript which is well-presented and well-argued.

What concentrations of capsaicin are achieved in vivo in rodents or in humans? Is a concentration of 200 micromolar likely to be in the therapeutic or toxic ranges? 

What are the likely effects of capsaicin on hepatocytes containing lipid droplets? Would capsaicin then be expected to reverse existing fatty liver or only to prevent lipid accumulation in normal hepatocytes?

Define AMPK in line 51, rather than line 60/61.

Author Response

Response to Reviewer 1 comments 

Point 1: What concentrations of capsaicin are achieved in vivo in rodents or in humans? Is a concentration of 200 micromolar likely to be in the therapeutic or toxic ranges?  

Capsaicin pharmacokinetic properties and bioavailability have been reviewed (1)Oral administration of capsaicin (30 mg/kg body weight for rats or 5 g capsaicin for humansachieve a maximal plasma concentration after 1 h administration equal to 2.47 µg/ml (2) in rats or 1.90 µg/m(3) in humanscorresponding to 8 µM and 6,2 µM, respectively. Plasma capsaicin concentration is far fewer of that used in our studyNevertheless, it should be note that the in vivo actions of capsaicin could be exerted by indirect mechanisms, such as activation the sympathetic nervous system or via hormonal secretion, as is the case of capsaicin-induced adiponectin secretionwhich could need lower capsaicin concentration than direct capsaicin actionsFor instance, capsaicin supra-pharmacological concentration (100 µM) has established to stimulate brown adipogenesis in vitro (4).  

Point 2: What are the likely effects of capsaicin on hepatocytes containing lipid droplets? Would capsaicin then be expected to reverse existing fatty liver or only to prevent lipid accumulation in normal hepatocytes? 

Our data provide in vitro evidence for an anti-lipogenic effect of capsaicinbut we have no evidences about the effect of capsaicin on hepatocytes, which represents a limitation of the study. However, other authors have demonstrated the ability of dietary capsaicin to prevent fatty liver accumulation (5) as well as to improve obesity-induced hepatic steatosis (6) in mice. This last study shows evidences concerning the capsaicin’s ability to reverse fatty liver. 

Point 3: Define AMPK in line 51, rather than line 60/61. 

We have defined AMPK in line 51, and we have revised the whole manuscript and corrected spelling and grammar errors.  

1.Rollyson WD, Stover CA, Brown KC, Perry HE, Stevenson CD, McNees CA, et al. Bioavailability of capsaicin and its implications for drug delivery. J Control Release. 2014;196:96-105. 

2.Suresh D, Srinivasan K. Tissue distribution & elimination of capsaicin, piperine & curcumin following oral intake in rats. Indian J Med Res. 2010;131:682-91. 

3.Chaiyasit K, Khovidhunkit W, Wittayalertpanya S. Pharmacokinetic and the effect of capsaicin in Capsicum frutescens on decreasing plasma glucose level. J Med Assoc Thai. 2009;92(1):108-13. 

4.Kida R, Noguchi T, Murakami M, Hashimoto O, Kawada T, Matsui T, et al. Supra-pharmacological concentration of capsaicin stimulates brown adipogenesis through induction of endoplasmic reticulum stress. Sci Rep. 2018;8(1):845. 

5.Li Q, Li L, Wang F, Chen J, Zhao Y, Wang P, et al. Dietary capsaicin prevents nonalcoholic fatty liver disease through transient receptor potential vanilloid 1-mediated peroxisome proliferator-activated receptor delta activation. Pflugers Arch. 2013;465(9):1303-16. 

6.Kang JH, Goto T, Han IS, Kawada T, Kim YM, Yu R. Dietary capsaicin reduces obesity-induced insulin resistance and hepatic steatosis in obese mice fed a high-fat diet. Obesity (Silver Spring). 2010;18(4):780-7. 

Reviewer 2 Report

In their manuscript, the authors studied effects of capsaicin on lipid metabolism related pathways using HepG2 cell line. The study is interesting and the manuscript is very well prepared.

Major points:

How did the authors choose the doses of caspsaicin? Did the authors check cellular viability following caspsaicin treatment at doses that were used in this study?

In Fig.4A, lysosome inhibitor co-treatment, e.g. chloroquine, is necessary to observe autophagy flux.

Author Response

Response to Reviewer 2 comments 

Major points:

Point 1: How did the authors choose the doses of capsaicin? Did the authors check cellular viability following capsaicin treatment at doses that were used in this study? 

In in vitro assays, capsaicin is usually administered at doses above 50 µM (1, 2)We selected 200 µM capsaicin since it was the minimal dose that decreased neutral lipid content in HepG2 cells under our experimental conditionsWe checked cell viability and results are published in Bort et al. 2019 (3)  

Point 2: In Fig.4A, lysosome inhibitor co-treatment, e.g. chloroquine, is necessary to observe autophagy flux. 

According to the reviewer’s recommendation, we have performed new experiments blocking autophagy flux with the lysosomal inhibitor chloroquineHepG2 cells were co-treated with 50 µM chloroquine (CQ) for 1 hour before the end of the assay. Results are shown in the new supplemental Figure S3Pharmacological inhibition of autophagy with CQ, which blocks the final step of autolysosomal degradation, resulted in enhancement of LC3 and p62 accumulation in both basal and capsaicin treated cells, consistent with autophagy flux inhibition. CQ treatment did not result in capsaicin-enhanced LC3II accumulation (maybe 8 hours of capsaicin treatment is a very short incubation time), whereas capsaicin significantly increased p62 accumulation. These results reinforce the notion that capsaicin induces autophagy blockage.  

1.Chen X, Tan M, Xie Z, Feng B, Zhao Z, Yang K, et al. Inhibiting ROS-STAT3-dependent autophagy enhanced capsaicin-induced apoptosis in human hepatocellular carcinoma cells. Free Radic Res. 2016;50(7):744-55. 

2.Zang Y, Fan L, Chen J, Huang R, Qin H. Improvement of Lipid and Glucose Metabolism by Capsiate in Palmitic Acid-Treated HepG2 Cells via Activation of the AMPK/SIRT1 Signaling Pathway. J Agric Food Chem. 2018;66(26):6772-81. 

3.Bort A, Sanchez BG, Spinola E, Mateos-Gomez PA, Rodriguez-Henche N, Diaz-Laviada I. The red pepper's spicy ingredient capsaicin activates AMPK in HepG2 cells through CaMKKbeta. PLoS One. 2019;14(1):e0211420. 

Round 2

Reviewer 2 Report

The manuscript has been significantly improved. It could be published in current version.